# Elucidating optical field directed hierarchical self-assembly of homogenous versus heterogeneous nanoclusters with femtosecond optical tweezers

Dipankar Mondal[1☯], Soumendra Nath Bandyopadhyay[1☯], Debabrata Goswami[1,2]*

**1** Department of Chemistry, Indian Institute of Technology Kanpur, Kanpur, India, **2** Center for Laser and Photonics, Indian Institute of Technology Kanpur, Kanpur, India

☯ These authors contributed equally to this work.
* dgoswami@iitk.ac.in

**Data Availability Statement:** All relevant data are within the manuscript and its Supporting Information files.

## Abstract

Insights into the morphology of nanoclusters would facilitate the design of nano-devices with improved optical, electrical, and magnetic responses. We have utilized optical gradient forces for the directed self-assembly of colloidal clusters using high-repetition-rate femtosecond laser pulses to delineate their structure and dynamics. We have ratified our experiments with theoretical models derived from the Langevin equation and defined the valid ranges of applicability. Our femtosecond optical tweezer-based technique characterizes the in-situ formation of hierarchical self-assembled clusters of homomers as well as heteromers by analyzing the back focal plane displacement signal. This technique is able to efficiently distinguish between nano-particles in heterogeneous clusters and is in accordance with our theory. Herein, we report results from our technique, and also develop a model to describe the mechanism of such processes where corner frequency changes. We show how the corner frequency changes enables us to recognize the structure and dynamics of the coagulation of colloidal homogeneous and heterogeneous clusters in condensed media over a broad range of nanoparticle sizes. The methods described here are advantageous, as the backscatter position-sensitive detection probes the in-situ self-assembly process while other light scattering approaches are leveraged for the characterization of isolated clusters.

## Introduction

The study of colloidal clusters at microscale dimensions provides insights into the process of self-organization for macromolecular agglomeration [1, 2]. These processes are also linked as well to the early stages of nucleation [3]. Cluster formation dynamics is of significance to the science of various natural processes, such as blood clotting, disease processes [4], protein crystallography [5], gelation [6], ceramics processing [7], etc. Furthermore, nanofabrication techniques based on the self-assembly dynamics of nanoparticles [8, 9] is important for the development of nanoscale components, devices, and systems in large quantities at lower costs.

**Funding:** Initial efforts of this work was started with some of the support that was still available from the Wellcome Trust; however, my Senior Fellow term has ended about 5-7 years back, so except the laser equipment, no further support was available from them. We thank the present support of the ISRO Science Technology Cell and SERB, Govt. of India. Dipankar Mondal and Soumendra Nath Bandyopadhyay thank UGC and CSIR, India, for their graduate fellowships.

**Competing interests:** The authors have declared that no competing interests exist.

Thus, understanding the self-assembly process can allow for control over the morphology and structures of nanoclusters [10]. The optical properties of these diverse nanostructures are highly sensitive to their geometry; thus, by exerting control over their structure, it is possible to engineer electric and magnetic responses over a broad range, which dictates the efficiency of the devices manufactured. Despite tremendous progress in directed assembly and self-assembly, a truly versatile assembly technique without specific functionalization of the colloidal particles remains elusive. Recently, optical tweezers [11] have inspired new experimental methodologies to investigate colloidal aggregation [12–15]. Here, we have used femtosecond optical tweezers to track the directed self-assembly of colloidal clusters due to the strong optical gradient field and show how it lends itself to novel insights into the existing assembly techniques.

Additionally, we have ratified our experimental results numerically and have predicted how monomers and their aggregation differ in terms of their Brownian dynamics by solving the Langevin equation for many-body systems. Combining experimental and theoretical methods, we have interpreted clusters size from corner frequency, as deduced by the power spectrum method [16]. The corner frequency is a characteristic frequency of a trapped particle that differs at every trapping laser power and is the limit beyond which the particle cannot remain trapped. Here, we have focused primarily on the development of a method for the enumeration of colloidal particles in hierarchical self-assembly processes via experimental techniques with theoretical support for a wide range of cluster sizes. Applying our techniques, we have characterized colloidal clusters by minimizing their mutual interactions. We note that there is still a small deviation between theoretical and experimental values mostly for particles of smaller sizes. This result indicates that there is a slight possibility of short-range and long-range interactions among the trapped particles that is unaccounted for. Although these interactions tend to hold the clusters in their most entropically favored structures [17], the optical gradient field is the dominant driving force, which dictates the structure of the clusters at the focal plane.

Apart from the single-particle force measurements, optical tweezers can be used as an intrinsic tool to infer the strength of the interactions between entities that form the clusters [18] by measuring the deviations from ideality. We expect that extensions to the semi-classical model will make it viable for the efficient profiling of in-situ intra- or inter-macromolecular interactions [19, 20] during the diffusing process. Optical trapping of multiple particles is observed in dense media such as dense colloidal suspensions or inside biological cells [21] due to the optical gradient forces generated by a tightly focused laser beam. Generally, the aggregation of colloidal particles is an irreversible process [22] but, optical field-directed colloidal clusters [23, 24] are reversible in nature, which we have confirmed through observation in our experimental results. Therefore, we can describe the different naturally occurring clusters without affecting the stable suspension state of the system as they transition easily from the trapped condition to the freely diffusing monomeric forms. Using a $TEM_{00}$ Gaussian beam, a harmonic potential is generated. A quadrant photodiode is used to monitor the position of the optically trapped object in this potential field by measuring the intensity fluctuations in the back focal plane [25, 26] of the objective that is perpendicular to the propagating beam. This method is crucial to the investigation of colloidal self-assembly processes and other fields of the colloidal sciences from self-replication [27] to aggregation-disaggregation transition [28]. During such processes, the local number concentration of colloids increases, which can be traced to the potential minima of the optical tweezers in order to decode their structural dynamics. In such a system, Brownian motion [29] is restricted, and corner frequency values are truncated, as observed in the frequency spectrum obtained from the power spectrum method.

We have trapped single beads as well as homodimers of 500 nm, 250 nm and 100 nm mean radius polystyrene beads to show that the working principle presented holds and extend the same principle to trimeric clusters consisting of 250 nm size particles within the limits of standard error. This self-assembly of identical spheres simplifies the analysis of the cluster assembly process and facilitates the fabrication of highly symmetric structures [10]. We have also trapped heterodimers of 500 nm, and 250 nm mean radius fluorophore coated polystyrene beads without surface modifications. The heterodimer formation process allows us to exert control over the exposed surface area of the nanoparticles and subsequently allows the control of their surface-active phenomenon [30]. Stable trapping of dimers is indicated by an increase in two-photon fluorescence in the live-feed video (S1 Video, S2 Video) and is further confirmed by the value of the corner frequency. The values of the corner frequency for each cluster are used to infer the number of particles present in the cluster and continuing the analysis during the heterodimer formation process indicates the size of the particles present in the heterodimer. Our method can easily be applied for the characterization of non-fluorescent non-interacting nanoclusters as well as for macromolecular assembly. Furthermore, we expect that our method will be applicable in the field of biosensing [31], nanoelectronics [32], surface-enhanced spectroscopies [33], nonlinear optics [34], etc.

## Experimental methods and materials

In our femtosecond optical tweezers (FOT) set up (Fig 1), the laser source was a mode-locked Ti-Sapphire laser (MIRA-900F pumped by Verdi-V5, Coherent Inc.), which generated femtosecond laser pulses at 13 ns separation centered at 780 nm wavelength. Our experimental measurements were taken at a pulse width of ~ 150 fs. A $\lambda/2$ waveplate in union with a polarizing beam splitter (PBS) controls the power of the input laser, which feeds into the optical tweezers setup that is fabricated in-house. The $\lambda/2$ waveplate is rotated, resulting in a rotation of the polarization of light. The fixed polarizing beam splitter allows a fraction of the laser power to pass according to its relative orientation with respect to the local axes. The two lenses, L1 and L2 work together as a beam-expander to ensure that the back aperture of the trapping objective is overfilled. The dichroic mirror, placed just after the lens assembly (L1-L2), reflects the 780 nm laser beam in the upward direction to the objective. A commercial oil immersion objective (UPlanSApo, 100X, 1.4 NA, OLYMPUS Inc. Japan) was used to focus trapping laser; simultaneously the forward scattered light from the trapped object was collected using another oil immersion objective (60x, PlanAapo N, 1.42 NA, OLYMPUS Inc. Japan). A dichroic mirror DM2 reflects this collimated beam which is focused by the lens L3 (if there is any fluorescence from the sample in the green region, it is cut off by the green filter, GF) on the quadrant photodiode QPD. The forward scattering data from the trapped particles were collected with a quadrant photodiode (QPD) (2901, Newport Co. USA) of rise time of 5 μs. This QPD output was connected to a digital oscilloscope (Waverunner 64Xi, LeCroy USA), which, in turn, interfaces with a personal computer through a GPIB card (National Instruments, USA). Data acquisition was done with the LABVIEW program and data analysis was performed with custom code run on MATLAB software suite. Two-photon fluorescence (TPF) from trapped particles was monitored using CCD camera (350 K pixel, e-Marks Inc. USA). White light is used for bright field illumination. The trapping laser power was measured with a power meter (FieldMate, Coherent USA) as well as a silicon amplified photodiode (PDA100A-EC, Thorlabs USA) before lens L1.

Commercially available polystyrene nanosphere solution with concentration $2.7 \times 10^{10}$ particles/ml was diluted in phosphate buffer solution (0.2 M phosphate buffer solution, pH = 7.4) and well sonicated for immediate use in trapping experiments. We have used 24×50 mm No. 0

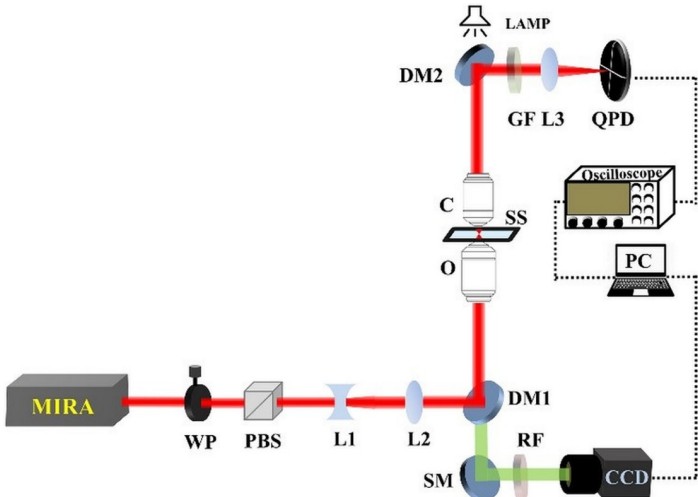

**Fig 1. Schematic diagram of our experimental femtosecond optical tweezers setup.** WP: Half-wave plate; PBS: Polarizing beam splitter; L1: Concave lens; L2: collimating convex lens; DM: Dichroic mirror; O: Objective lens; SS: Sample stage; C: Condenser lens; GF: Green filter; L3: Focusing lens; QPD: Quadrant photodiode; SM: Silver mirror; RF: Red filter; CCD: Camera (Charge-coupled device) PC: personal computer.

cover glass with an assembled by placing a coverslip 22×22 mm No. 1 separated by spacers of double-sided sticky tape. The sizes of the particles are confirmed through dynamic light scattering measurements (Fig 2).

## Results and discussions

We have used a 780 nm pulse laser to trap a single polystyrene bead and have tracked their self-assembly in dense media. The low power trapping laser has a nominal heating effect because of the very low absorption coefficient [35, 36] of the solution used at 780 nm. We have used oil immersion objective and the refractive index of the oil used is 1.518. The objective lens has a transmission of ~70% at 780 nm wavelength and a working distance of 100 μm.

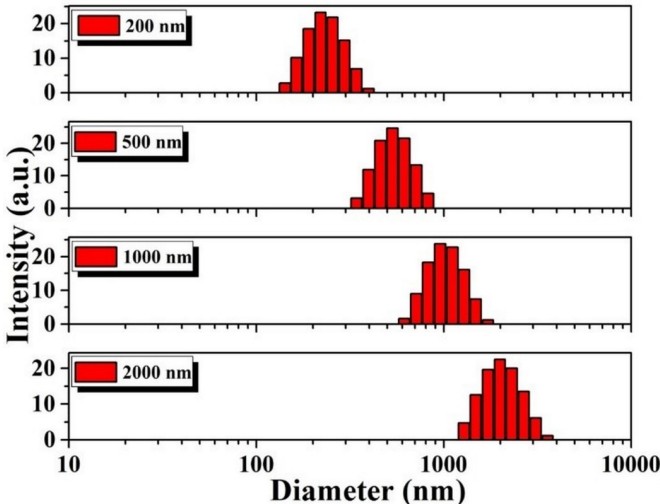

**Fig 2. Size distribution of the polystyrene beads measured by Dynamics light scattering.**

When a micron size particle in a viscous [37] Newtonian fluid exhibits Brownian motion under the influence of an oscillating harmonic potential well, the equation of motion for such a particle can be expressed by the following form of the Langevin equation [38, 39] (under the approximation that the motion of micrometer-sized particles takes place at a small Reynolds number where viscous drag dominates inertial forces) as seen in Eq 1:

$$\gamma \dot{x}(t) + \kappa x(t) = \zeta_{therm}(t) \tag{1}$$

Where x(t) is time dependent position, $\gamma$ is the viscous drag coefficient as per Stokes' Law, $\kappa$ is spring constant and $\zeta_{therm} = \sqrt{(2k_B T\gamma)}F(t) = \gamma\sqrt{(2D)}F(t)$ is the time dependent random thermal force. The diffusion coefficient can be expressed by Einstein equation $D = \frac{k_B T}{\gamma}$; where $k_B$ is Boltzmann constant and $T$ is experimental room temperature. By solving the above equation, we can fit our experimental one-sided power spectrum ($P_x(f)$) to a theoretical power spectrum [16, 40] given by Eq 2:

$$
\left.
\begin{aligned}
P_x(f) &= \frac{1}{T_{msr}}\left(|\tilde{x}(f)|^2 + |\tilde{x}(-f)|^2\right) \quad 0 \le f \le \frac{T_{msr}}{2} \\
&= \frac{2}{T_{msr}}|\tilde{x}(f)|^2 \\
&= \frac{D}{\pi^2(f_c^2 + f^2)} \\
\text{Where, } \tilde{x}_f(f) &= \int_{-T_{msr}/2}^{T_{msr}/2} dt\, e^{i2\pi f_k}\, x(t),\ f_k \equiv \frac{k}{T_{msr}},\ k \text{ is integer}
\end{aligned}
\right\} \tag{2}
$$

Here, $T_{msr}$ is the measurement time, $f$ is the frequency, and $\tilde{x}_f(f)$ is the Fourier transform of the time dependent position given by $x(t)$. In a colloidal suspension, multiple particle trapping has been observed as well [12]. For the trapping of two identical colloidal particles, the time-averaged power spectrum is determined from the following equation:

$$
\left.
\begin{aligned}
P_{2p_I}(f) &= \frac{D}{\pi^2(f_c^2 + f^2)} + \frac{D}{\pi^2(f_c^2 + f^2)} \\
&= \frac{2D}{\pi^2(f_c^2 + f^2)} \\
&= \frac{D}{\pi^2\left(\left(f_c/\sqrt{2}\right)^2 + \left(f/\sqrt{2}\right)^2\right)} = \frac{D}{\pi^2\left(\left(f_c/\sqrt{2}\right)^2 + f^2\right)}
\end{aligned}
\right\} \tag{3}
$$

Since $f$ is an $x$ axis dummy variable, therefore, on dividing the axis by the square root of two, it transforms into a new $x$ coordinate, which we denote by the same variable, $f$. The diffusion coefficient of the cluster is D. A more generalized formula for a cluster of homogenous $N$ number of particles is thus:

$$P_{NI}(f) = \frac{D}{\pi^2\left(\left(f_c/\sqrt{N}\right)^2 + \left(f/\sqrt{N}\right)^2\right)} = \frac{D}{\pi^2\left(\left(f_c/\sqrt{N}\right)^2 + f^2\right)} \tag{4}$$

We have analyzed the position fluctuation data of the trapped beads collected using QPD [41] at a 100 kHz sampling rate over the duration of 2.5 seconds. The data acquired from the X and Y channels are de-correlated by removing the cross-talk [42] between these channels and the power spectrum is subsequently obtained using MATLAB code.

Optical tweezers can capture nanospheres using optical forces but lack the capability to perform dynamic manipulation [43]. Here, we have reported the reversible assembly of nanoparticles using femtosecond optical tweezers; the optically driven reversible clusters are shown in Fig 3A and is also shown in the supporting videos (S1 Video, S2 Video) as the sequential trapping events clearly show reversibility of the process. For a better understanding of the trapping event, we have divided the raw data into different regions (shown as 1–5 inside red circles of Fig 3). The region denoted by 1 inside the red circle indicates that there is no particle inside the foal volume. Power spectrum analysis of this region, corresponding to Fig 3B results in the light null spectrum which has a low-frequency slope with -1.71 $V^2.Hz^{-2}$. The area denoted by 2 is a long spike in QPD signal, which is observed as a consequence of biased diffusion of the particle into the focal region [44].

The region marked 3 (Fig 3) is when the first particle is stably trapped inside the focus, the analysis of which yield a corner frequency value of 111 Hz. When a second particle is attracted by the highly focused laser beam, trapping is observed at around 8 second. This denotes the starting of region 4 where the dimer forms and is stably trapped. This event can be easily identified by the sudden increase in voltage output from the QPD. This increase in voltage is due to the large effective size of the cluster, which scatters light to a greater extent and indicates greater displacement. The diffusion coefficient is inversely proportional to the effective radius of the diffusing particle at constant temperature [45, 46]. We verify this experimentally and observe this trend in the data generated in the analysis of the clusters as well. In this region, the corner frequency is 70 Hz. According to our theoretical calculations, 111 Hz ($f_c$ of the single particle in this experimental condition) will be reduced to 78.5 Hz ($\frac{111}{\sqrt{2}} = 78.5$) if another exactly identical particle gets trapped in the presence of the first particle.

Experimentally, the small deviation in corner frequency may arise due to the fact that the particle sizes vary within limits given by 500 ± 50 nm and are thus not exactly identical. However, this effect may be the result of subtle interactions between the trapped particles, though such interactions are suppressed and negligible over the range of laser powers used. Subsequent trapping regions are of great interest as they prove the reversibility of the cluster formation dynamics due to the optical trapping gradient. Region 5 begins at the 13 second-mark and denotes where one particle has escaped the trap while the other particle remains stably inside the trap for a long time. This also supports our assertion that the two particles have almost negligible interactions inside the laser-mediated cluster and either one of the two particles do not disturb the other trapped particle while exiting the trap. This region of the trap has a corner frequency of 107 Hz. We note that it is difficult to determine whether the initially trapped particle remains up to region 5 or if the second incoming particle is the one which remains in the trap alone after a particle exits at the end of region 4.

Furthermore, to validate the generality of the Eq 4 for different trapping powers, we have performed a power dependent study of the cluster formation of 500 nm radius microspheres. This shows that at low powers (5 mW to 25 mW average laser powers) the particles can be considered to be non-interacting and the theoretical equation derived previously, Eq 4 holds. However, at higher power (30–35 mW), deviations are observed, and at even higher powers stable trapping is not observed for even single particles. We have also calculated the trap stiffness (κ) for monomer and dimers (Table 1) and observed that dimers have higher stiffness values compared to monomers, as expected.

Thus, we have reported here the domain determined by low laser power limits where our assumptions for the inter-particle negligible interactions hold (Fig 4 and Table 1). The power dependent corner frequency of a singly trapped microsphere has a slope of $(df_C/dP)_{1P} = 6.5$ Hz/mW, whereas the doubly trapped microsphere has a slope of $(df_C/dP)_{2P} = 4.1$ Hz/mW (Fig

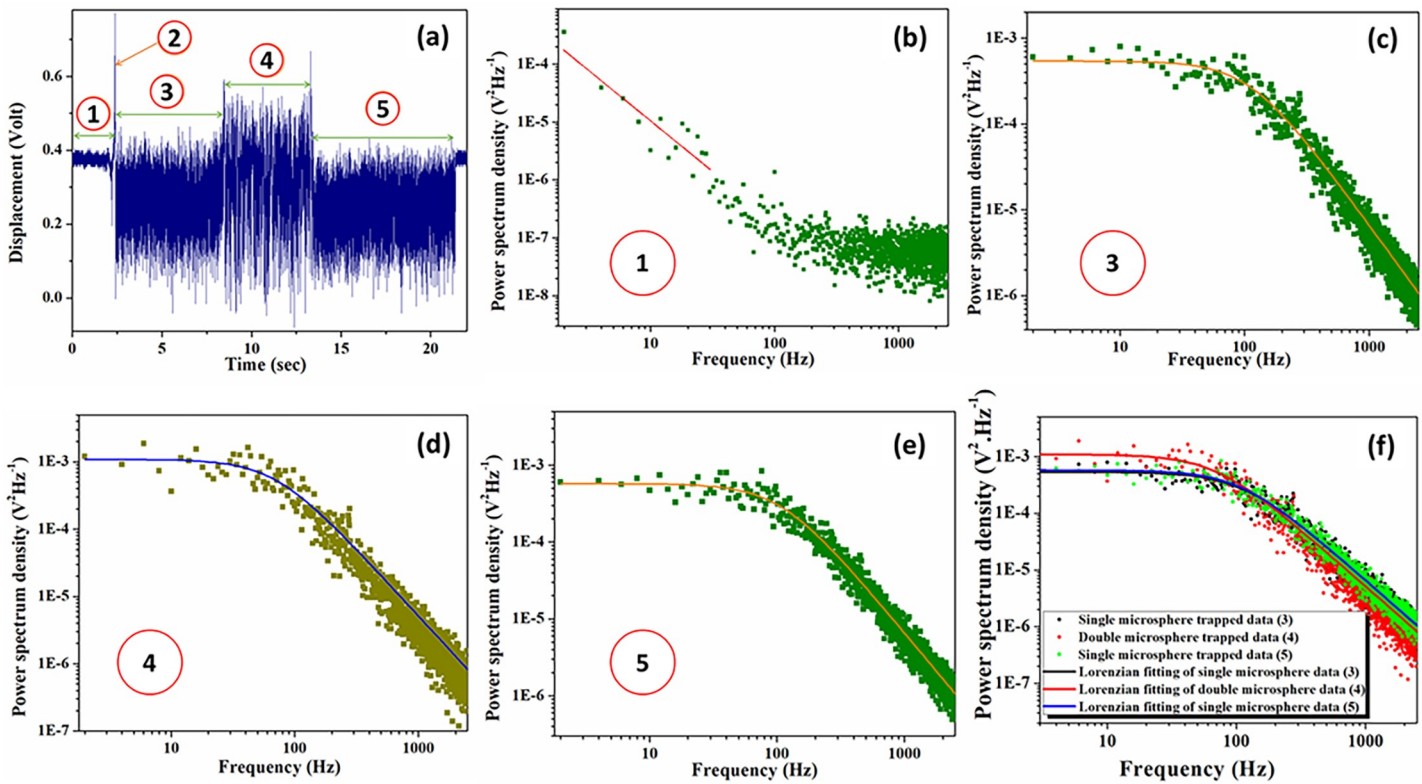

**Fig 3. Typical raw data (at 17.5 mW average laser power) and its corresponding fitted data for a reversible dimer formation event in the sequence.** (a) Experimentally measured trapped bead displacement raw data showing different regions of trapping event. (b) Light null spectrum: Power spectrum when no microsphere is trapped but trapping light imposing onto QPD. (c) Power spectrum of the region 3 from the raw data (green square) and its Lorentzian fitting (orange line) gives $f_C$ = 111 Hz for a single trapped particle. (d) The power spectrum of region 4 of the raw data (olive square) and its Lorentzian fitting (blue line) gives $f_C$ = 70 Hz corresponding to the dimer. (e) Power spectrum of mark 5 of the raw data (green square) and its Lorentzian fitting (orange line) gives $f_C$ = 107 Hz. (f) Power spectrum showing the power spectrum of region 3, 4 and 5 altogether for comparison purpose.

4G). The ratio of single and double particle corner frequencies is theoretically predicted, to be $\sqrt{2} = 1.414$. However, our experimentally observed ratio is 1.585. As said previously the nearly 8% error may be coming due to the fact that each particle is not identical in size as well as there might be little inter-particle interaction within the clusters inside the trap.

We have also shown that our equation (Eq 8) holds true even on varying the particle size in the range of 500 nm to 250 nm, and also to 100 nm range. This supports the generality of the equation and its insensitivity to the particle size.

**Table 1. Corner frequency and trap stiffness of the monomers and dimers, under the optically tweezed condition, for 500 nm radius polystyrene beads in water.**

| Trapping laser power (mW) | Corner frequency of single microsphere (Hz) | Corner frequency of double microsphere (Hz) | Trap stiffness of single microsphere (fN/nm) | Trap stiffness of double microsphere (fN/nm) |
|---|---|---|---|---|
| 5 | 26 | 16 | 1.37 | 1.68 |
| 10 | 69 | 41 | 3.63 | 4.32 |
| 15 | 99 | 61 | 5.21 | 6.42 |
| 17.5 | 109 | 70 | 5.74 | 7.37 |
| 20 | 129 | 84 | 6.80 | 8.85 |
| 25 | 164 | 105 | 8.64 | 11.06 |

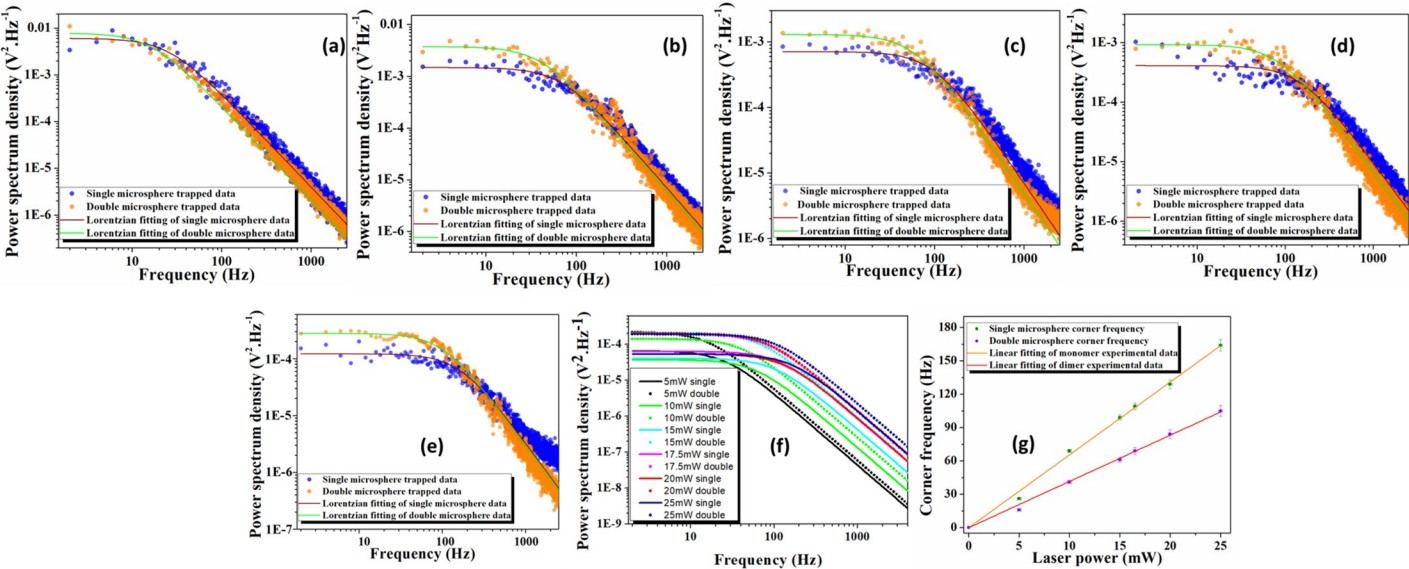

**Fig 4. Experiments, fits and comparisons** (a-e) The experimentally measured one-sided power spectrum (blue sold circle for single microsphere and orange sold circle for double microsphere) and the respective Lorentzian fit to data (brown line for single microsphere and green line for double microsphere) for 500 nm radius fluorophore coated polystyrene bead at increasing laser powers. (f) The experimentally measured all fitted one-sided power spectrum of 500 nm radius polystyrene beads merged in a same plot (g) The comparison between single and double polystyrene microsphere.

We have performed a power dependent study for 250 nm radius beads. We have analyzed their corner frequency when trapped as a monomer, dimer and trimer (Fig 5A). We have deduced a slope of 12.8 Hz/mW for the monomer. The slope observed for a doubly trapped bead is 8.7 Hz/mW (Fig 5B). The ratio of these corner frequencies is 1.471 (which is slightly higher than the theoretical prediction of $\sqrt{2}$). Thus, we note that the theoretically predicted dimer is 9.1 Hz/mW, which is higher than the experimentally observed value, but is in keeping with the observed ratio. The power dependent fitted lines of monomers, dimers and trimers

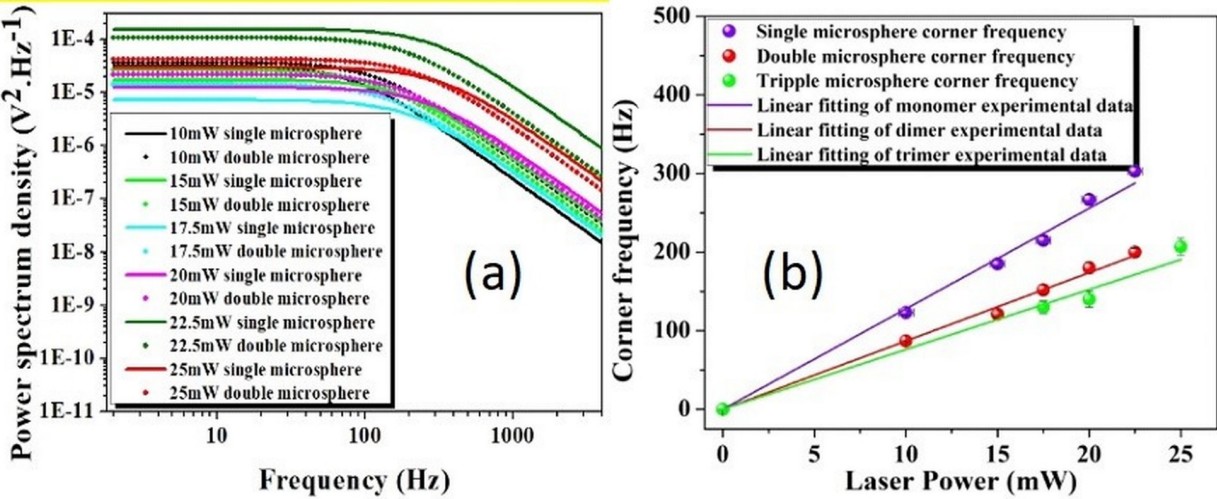

**Fig 5. Power spectrum and corner frequency:** (a) The experimentally measured fitted one-sided power spectrum of 250 nm radius polystyrene beads under varying gradient fields, and (b) power dependent corner frequency of monomer, dimer and trimer of 250 nm radius trapped bed.

**Table 2. Corner frequency of the monomer, dimer and trimer, under the optically tweezed condition, for 250 nm particle at different laser powers.**

| Trapping laser power (mW) | Corner frequency of single microsphere (Hz) | Corner frequency of double microsphere (Hz) | Corner frequency of triply trapped microsphere (Hz) |
|---|---|---|---|
| 10 | 123 | 87 | |
| 15 | 185 | 121 | |
| 17.5 | 215 | 152 | 124 |
| 20 | 267 | 180 | 140 |
| 22.5 | 303 | 200 | |
| 25 | 343 | 238 | 207 |

considered are given in Table 2. Our results validate the high sensitivity of our measurement techniques.

We have also trapped cluster of three particles, each of radius 250 nm, and these follow the expected trend ($f_C/N^{1/2}$), but within a shorter power range. This is because of the fact that, at very low powers, the potential inside the trap is unable to hold more than two particles simultaneously. As a result, the third particle does not get trapped. Again, at higher power, the particles interact non-negligibly and are expected to collide with each other, thus overcoming the trapping position and allowing for escape from the trap. We note that the sample preparation was complicated by competing considerations for convergence with our theoretical model. The colloidal solution must be dilute enough to ensure negligible inter-particle interaction in the solution. At the same time, the particle density should be high enough such that the probability of trapping the third particle is not unfeasible. Hence there is a competitive trade-off between these two factors. For the smallest (100 nm) particle, we have used a 20% aqueous glycerin solution which has a higher viscosity [47, 48] so as to make the trap stable over a longer time period as the power-spectrum method works better over longer trapping times [49] for the minimization of errors. Similarly, 100 nm radius beads show a ratio of 1.31 between the power dependent fitted lines of singly and doubly trapped particles (Fig 6). The corresponding data have been reported in Table 3.

For further confirmation of our methods, we have verified our results for systems consisting of a mixture of 500 nm and 250 nm mean radius particles (Fig 7A) and observed their power dependent behavior (Fig 7B).

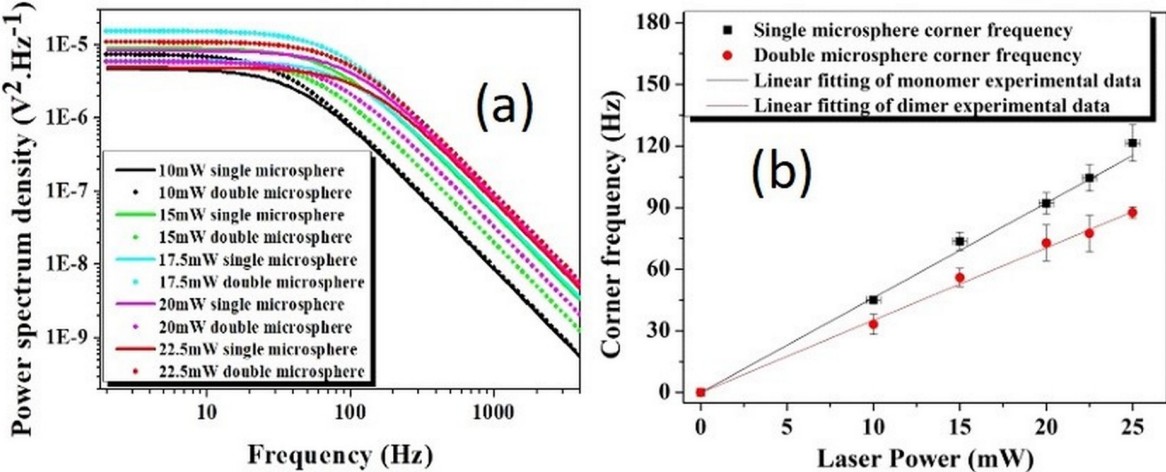

**Fig 6. The experimentally measured data** depicting (a) the fitted one-sided power spectrum of 100 nm radius polystyrene beads under different gradient field and (b) power dependent corner frequency of monomer and dimer of 100 nm radius trapped bed.

**Table 3. Corner frequency of the monomers and dimers, under the optically tweezed condition, for 100 nm nanosphere at different laser powers.**

| Trapping laser power (mW) | Corner frequency of single microsphere (Hz) | Corner frequency of double microsphere (Hz) |
|---|---|---|
| 10 | 45 | 33 |
| 15 | 74 | 56 |
| 17.5 | 92 | 73 |
| 20 | 105 | 77 |
| 22.5 | 122 | 88 |

We have also solved the Langevin equation for two non-identical and non-interacting trapped particles, and the power spectral density can be represented by the following equation:

$$P_{2p_{NI}}(f) = \frac{D_1}{\pi^2(f_{c_1}^2 + f^2)} + \frac{D_2}{\pi^2(f_{c_2}^2 + f^2)}$$

$$P_{2p_{NI}}(f) = \frac{(D_1 f_{c_2}^2 + D_1 f^2 + D_2 f_{c_1}^2 + D_2 f^2)}{\pi^2(f_{c_1}^2 + f^2)(f_{c_2}^2 + f^2)} = D_{12}\frac{(A^2 + 2f^2)}{\pi^2(f_{c_1}^2 + f^2)(f_{c_2}^2 + f^2)} \tag{5}$$

Where, $A = \sqrt{f_{c_1}^2 + f_{c_2}^2}$ and the above equation is valid under the approximation: $D_1\ and\ D_2 \approx \frac{(D_1 + D_2)}{2} = D_{12}$. This is due to the fact that the overall diffusion coefficient is seen as the cluster forms and the mass of trapped particles increases. Using Eq 5 we have fitted the fluctuations of the heterodimer data at different powers. We have observed optically induced gradient fields, which match our theoretically predicted value through Eq 5. At high peak powers, smaller size particles exhibit non-linear phenomena [50] and consequently the experimental and theoretical values deviate. The fitted data or the heterodimers are given in Table 4.

At powers less than 5 mW, multi-particle optical trapping was not observed. This may be attributed to the fact that the potentials generated at such low laser powers are not sufficient to trap more than one particle. At an average power higher than 30 mW, the experimental corner frequency values tend to deviate from the given theory, which may be due to the increased interaction among the trapped particles within the small focal volume and needs further investigation.

We believe from our experimental results that our proposed equation we have derived (Eq 5) can be used to characterize the heterogeneous clusters without further modification of the colloid spheres. This, in turn, is able to quantify the structure and function of globular systems like micelles, vesicles, plasmids etc. Our methods are accurate up to third decimal point as demonstrated in our analysis and can be applied for sensitive biosensing [51]. Furthermore, we believe that this technique can also be utilized for controlled hetero-aggregation [52].

## Conclusions

The anomalous behavior of single particle and aggregate cluster diffusion has been investigated experimentally, and a theoretical model for the same has been developed. Our method, based on the corner frequency analysis of power spectrum data, is able to probe the Brownian motion of identical and non-identical colloidal hierarchical self-assembly in a mixture of different nano-particles. The experimental results obtained from our femtosecond optical tweezer technique have also been ratified theoretically. Our theory and experiments account for the fact that the inter-particle interactions must be minimized while the trapping probability

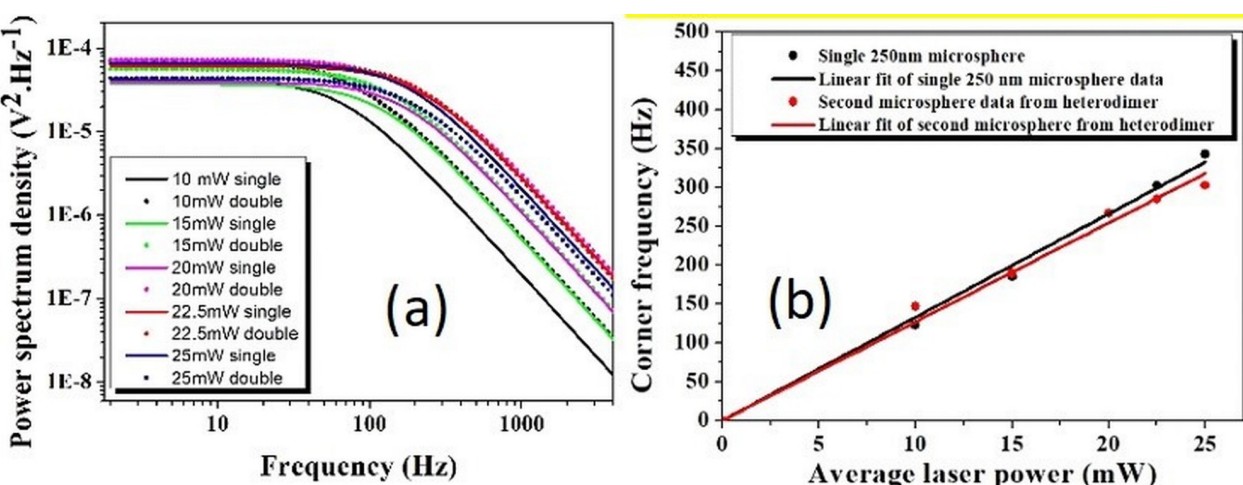

**Fig 7. The experimentally measured one-sided power spectrum of monomers and heterodimers** of (a) 500 nm and 250 nm radius polystyrene beads under different gradient fields (b) power dependent corner frequency of monomer and dimer of 250 nm and 500 nm radius heterodimer trapped bead.

**Table 4. Corner frequency of heterodimers formed from 100 nm and 250 nm nanospheres, at different laser powers and under the optically tweezed condition, at different laser powers.**

| Trapping laser power (mW) | Corner frequency of single microsphere (Hz) | Corner frequency of double microsphere (Hz) |
|---|---|---|
| 10 | 68 | 147 |
| 15 | 109 | 190 |
| 20 | 130 | 267 |
| 22.5 | 164 | 285 |
| 25 | 180 | 303 |

should be favorable. This method can be further utilized for the characterization of cluster size and is able to track the size of particles coming into the cluster during the sequential trapping phenomena. We have also shown that the clusters formed under the influence of optical gradients are reversible. Our theoretical model investigation works well for the cluster, which remain within the focal volume or when the size is nearly the same size as the focal volume. We have investigated the effect of agglomeration on the Brownian motion of the system and found that higher mass leads to a decrease in the corner frequency due to the restricted Brownian motion. The theory is shown to be valid for the range of experimental data within the power range of our study and our methodology is applicable to a wider range of studies pertaining to hetero-aggregation and other aspects of rheology. Our methodology is thus viable for sensitive biosensing diverse applications from tracking microrheological changes inside single cells to detecting aging of crowding due to fibrillation inside living cells.

## Supporting information

**S1 Video. 100 nm radius multi-particle trapping process.**
(AVI)

**S2 Video. 500 nm radius multi-particle trapping process.**
(AVI)

## Acknowledgments

We thank Rohit Goswami and Sonaly Goswami of the Femtolab at IIT Kanpur for changes to the linguistic style and formatting. Amrita Goswami of the Chemical Engineering department also assisted with the copy-editing.

## Author Contributions

**Conceptualization:** Soumendra Nath Bandyopadhyay, Debabrata Goswami.

**Data curation:** Dipankar Mondal.

**Formal analysis:** Dipankar Mondal, Soumendra Nath Bandyopadhyay.

**Investigation:** Soumendra Nath Bandyopadhyay.

**Methodology:** Dipankar Mondal.

**Project administration:** Debabrata Goswami.

**Software:** Dipankar Mondal.

**Supervision:** Debabrata Goswami.

**Validation:** Soumendra Nath Bandyopadhyay.

**Visualization:** Dipankar Mondal, Debabrata Goswami.

**Writing – original draft:** Dipankar Mondal.

**Writing – review & editing:** Soumendra Nath Bandyopadhyay, Debabrata Goswami.

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
