## [Decision Letter · Decision Letter 0]

4 Jul 2019

PONE-D-19-15382

Elucidating Optical Field Directed Hierarchical Self-assembly of Homogenous versus Heterogenous Nano Clusters with femtosecond optical tweezers

PLOS ONE

Dear Dr. Goswami,

Thank you for submitting your manuscript to PLOS ONE. After careful consideration, we feel that it has merit but does not fully meet PLOS ONE’s publication criteria as it currently stands. Therefore, we invite you to submit a revised version of the manuscript that addresses the points raised during the review process.

We would appreciate receiving your revised manuscript by Aug 18 2019 11:59PM. To enhance the reproducibility of your results, we recommend that if applicable you deposit your laboratory protocols in protocols.io, where a protocol can be assigned its own identifier (DOI) such that it can be cited independently in the future. For instructions see: http://journals.plos.org/plosone/s/submission-guidelines#loc-laboratory-protocols

We look forward to receiving your revised manuscript.

Kind regards,

Fang-Bao Tian

Academic Editor

PLOS ONE

Journal Requirements:

Additional Editor Comments (if provided):

Reviewers' comments:

Reviewer's Responses to Questions

**Comments to the Author**

1. Is the manuscript technically sound, and do the data support the conclusions?

Reviewer #1: Yes

Reviewer #2: Partly

2. Has the statistical analysis been performed appropriately and rigorously? 

Reviewer #1: Yes

Reviewer #2: Yes

3. Have the authors made all data underlying the findings in their manuscript fully available?

Reviewer #1: Yes

Reviewer #2: Yes

4. Is the manuscript presented in an intelligible fashion and written in standard English?

Reviewer #1: Yes

Reviewer #2: Yes

5. Review Comments to the Author

Reviewer #1: First of all, the mechanism of the experiment is not clearly stated. How does the laser trapping reveals the structures and distribution of the polystyrene beads? What is collected after QPD and what is analyzed in the computer? It also didn't cover how the laser is going through each experiment instruments and how it works afterwards. In addition, it mentioned the PBS: Polarizing beam splitter, where PBS is also used in "Commercially available polystyrene nanosphere solution with concentration 2.7×1010 particles/ml was diluted in phosphate buffer solution (0.2 M PBS, PH=7.4) and well sonicated for immediate use in trapping experiments." Are them the same?

Second, in the beginning of the section of RESULTS AND DISCUSSION, it mentioned "The low power trapping laser has a nominal heating effect

because of the very low absorption coefficient of solution used at 780 nm.". If that's the case, how would this heating effect affect the trapping experiment and measurements?

Third, the equations derived are not clear and may contain issues. How is equation 2 being derived to the final form, where the position related term is eliminated? Also what is the f_c? And how is equation 3 being derived to final form, where f_c has a coefficient of one over square root of 2 and f has no coefficient? From what it presents, f_c and f are supposed to be having the same coefficient. Similar issue is related to equation 5. All these equations require more detailed derivation and checking.

Fourth, what is the trapping stiffness showing in table I? I don't see any term related to that.

Reviewer #2: Predication of morphology is very important to design and facilitate materials with high performance properties. The authors stated they developed a theoretical method to predict morphologies of nano-clusters by solving the Langevin equation. It is a meaningful topic of the nano-research, and the present paper would have a wide interest of readers. It is suitable for the journal, PLOS ONE. However, there are several key problems to be clarified before accepting.

1. Section Results and Discussion, the authors list Langevin equation (i.e. equation 1), and the equation derived from Langevin equation. They stated “By solving the above equation, we can fit our experimental one-sided power spectrum into theoretical power spectrum”. It lacks of necessary process for derivation. Because the authors stated they provide a model, and equation 2 is the governing equation of the model. It at least refers to the model. Please provide more details on the derivation from equation 1 to 2.

2. In tables 1, 2 and 3, Why did the authors choose the values of the trapping laser power which are not the arithmetic sequences. Why are there the special values of 17.5, 22.5?

3. The text below figure 4, lines 1-3, “To further investigation into this”, what does “this” refer to? “This shows that at low power”, what is the mean of “this”? “the particles can be considered as non-interacting, and the theoretical equation derived before holds.” How could the “non-interacting” be concluded? Please explain more about it.

4. All the equations, please explain the physical meanings of every symbol. I failed to find the exact explanations of P_x, x^{\\tilde}, f_c.

5. Section Conclusion, line 2 from bottom, what does the “fluctuation-dissipation theorem” refer to in the present paper? Does it refer to Langevin equation or the equations corresponding to Langevin equation? The reviewer think that the data partly support such the conclusion.

6. Line 3 below equation 1, kB should be k_B.

6. PLOS authors have the option to publish the peer review history of their article (what does this mean?). If published, this will include your full peer review and any attached files.

Reviewer #1: No

Reviewer #2: No

---

## [Author Response · Author response to Decision Letter 0]

15 Aug 2019

All issued raised by reviewers has been addressed and response to each of the reviewers are attached as files.

---

## [Decision Letter · Decision Letter 1]

9 Sep 2019

PONE-D-19-15382R1

Elucidating Optical Field Directed Hierarchical Self-assembly of Homogenous versus Heterogenous Nano Clusters with femtosecond optical tweezers

PLOS ONE

Dear Dr. Goswami,

Thank you for submitting your manuscript to PLOS ONE. After careful consideration, we feel that it has merit but does not fully meet PLOS ONE’s publication criteria as it currently stands. Therefore, we invite you to submit a revised version of the manuscript that addresses the points raised during the review process.

We would appreciate receiving your revised manuscript by Oct 24 2019 11:59PM. To enhance the reproducibility of your results, we recommend that if applicable you deposit your laboratory protocols in protocols.io, where a protocol can be assigned its own identifier (DOI) such that it can be cited independently in the future. For instructions see: http://journals.plos.org/plosone/s/submission-guidelines#loc-laboratory-protocols

We look forward to receiving your revised manuscript.

Kind regards,

Fang-Bao Tian

Academic Editor

PLOS ONE

Additional Editor Comments (if provided):

A minor correction is required as suggested by Review 1.

Reviewers' comments:

Reviewer's Responses to Questions

**Comments to the Author**

1. If the authors have adequately addressed your comments raised in a previous round of review and you feel that this manuscript is now acceptable for publication, you may indicate that here to bypass the “Comments to the Author” section, enter your conflict of interest statement in the “Confidential to Editor” section, and submit your "Accept" recommendation.

Reviewer #1: All comments have been addressed

Reviewer #2: All comments have been addressed

2. Is the manuscript technically sound, and do the data support the conclusions?

Reviewer #1: Yes

Reviewer #2: Yes

3. Has the statistical analysis been performed appropriately and rigorously? 

Reviewer #1: Yes

Reviewer #2: Yes

4. Have the authors made all data underlying the findings in their manuscript fully available?

Reviewer #1: Yes

Reviewer #2: Yes

5. Is the manuscript presented in an intelligible fashion and written in standard English?

Reviewer #1: Yes

Reviewer #2: Yes

6. Review Comments to the Author

Reviewer #1: The author has provided detailed replies to all the comments. I am not sure if it is some conversion error. Equation 3 and 4 are still not showing in the correct form, while they were in correct form in the section of response to the comments. I urge the author to fix the appropriate typo errors.

Reviewer #2: The authors have made a considerable improvement to meet the queries raised by the referees. Hence I suggest accepting the manuscript for publication.

7. PLOS authors have the option to publish the peer review history of their article (what does this mean?). If published, this will include your full peer review and any attached files.

Reviewer #1: No

Reviewer #2: No

---

## [Author Response · Author response to Decision Letter 1]

23 Sep 2019

We are very happy to note that both the reviewers have accepted our changes. The first reviewer has noted the file conversion error for Word to PDH and we are correcting that through this revision. Thank you.

---

## [Decision Letter · Decision Letter 2]

26 Sep 2019

Elucidating Optical Field Directed Hierarchical Self-assembly of Homogenous versus Heterogenous Nano Clusters with femtosecond optical tweezers

PONE-D-19-15382R2

Dear Dr. Goswami,

We are pleased to inform you that your manuscript has been judged scientifically suitable for publication and will be formally accepted for publication once it complies with all outstanding technical requirements.

With kind regards,

Fang-Bao Tian

Academic Editor

PLOS ONE

Additional Editor Comments (optional):

Reviewers' comments:

Reviewer's Responses to Questions

**Comments to the Author**

1. If the authors have adequately addressed your comments raised in a previous round of review and you feel that this manuscript is now acceptable for publication, you may indicate that here to bypass the “Comments to the Author” section, enter your conflict of interest statement in the “Confidential to Editor” section, and submit your "Accept" recommendation.

Reviewer #2: All comments have been addressed

2. Is the manuscript technically sound, and do the data support the conclusions?

Reviewer #2: Yes

3. Has the statistical analysis been performed appropriately and rigorously? 

Reviewer #2: Yes

4. Have the authors made all data underlying the findings in their manuscript fully available?

Reviewer #2: Yes

5. Is the manuscript presented in an intelligible fashion and written in standard English?

Reviewer #2: Yes

6. Review Comments to the Author

Reviewer #2: (No Response)

7. PLOS authors have the option to publish the peer review history of their article (what does this mean?). If published, this will include your full peer review and any attached files.

Reviewer #2: No

---

## [Editor Report · Acceptance letter]

15 Oct 2019

PONE-D-19-15382R2 

Elucidating Optical Field Directed Hierarchical Self-assembly of Homogenous versus Heterogenous Nano Clusters with femtosecond optical tweezers 

Dear Dr. Goswami:

I am pleased to inform you that your manuscript has been deemed suitable for publication in PLOS ONE. Congratulations! Your manuscript is now with our production department. 

With kind regards,

on behalf of

Dr. Fang-Bao Tian 

Academic Editor

PLOS ONE